ecology, plant science

cattle, fire, grazing, grassland, functional traits, megafauna

**Author for correspondence:**
Cédrique L. Solofondranohatra
e-mail: lovacedrique@gmail.com

# Fire and grazing determined grasslands of central Madagascar represent ancient assemblages

Cédrique L. Solofondranohatra[1,2], Maria S. Vorontsova[3], Gareth P. Hempson[4], Jan Hackel[3], Stuart Cable[2,5], Jeannoda Vololoniaina[1] and Caroline E. R. Lehmann[4,6,7]

[1]Laboratoire de Botanique, Département de Biologie et Ecologie Végétales, Faculté des Sciences, Université d'Antananarivo, Antananarivo, Madagascar
[2]Kew Madagascar Conservation Centre, Antananarivo, Madagascar
[3]Comparative Plant and Fungal Biology, Royal Botanic Gardens Kew, London, UK
[4]Centre for African Ecology, School of Animal and Plant Sciences, University of the Witwatersrand, Johannesburg, South Africa
[5]Conservation Science, Royal Botanic Gardens Kew, London, UK
[6]School of GeoSciences, The University of Edinburgh, Edinburgh, UK
[7]Tropical Diversity, Royal Botanic Garden Edinburgh, Edinburgh, UK

CLS, 0000-0003-1808-4954; MSV, 0000-0003-0899-1120; GPH, 0000-0001-8055-4895; JH, 0000-0002-9657-5372; SC, 0000-0001-7119-9844; JV, 0000-0002-5178-6889; CERL, 0000-0002-6825-124X

The ecology of Madagascar's grasslands is under-investigated and the dearth of ecological understanding of how disturbance by fire and grazing shapes these grasslands stems from a perception that disturbance shaped Malagasy grasslands only after human arrival. However, worldwide, fire and grazing shape tropical grasslands over ecological and evolutionary timescales, and it is curious Madagascar should be a global anomaly. We examined the functional and community ecology of Madagascar's grasslands across 71 communities in the Central Highlands. Combining multivariate abundance models of community composition and clustering of grass functional traits, we identified distinct grass assemblages each shaped by fire or grazing. The fire-maintained assemblage is primarily composed of tall caespitose species with narrow leaves and low bulk density. By contrast, the grazer-maintained assemblage is characterized by mat-forming, high bulk density grasses with wide leaves. Within each assemblage, levels of endemism, diversity and grass ages support these as ancient assemblages. Grazer-dependent grasses can only have co-evolved with a now-extinct megafauna. Ironically, the human introduction of cattle probably introduced a megafaunal substitute facilitating modern day persistence of a grazer-maintained grass assemblage in an otherwise defaunated landscape, where these landscapes now support the livelihoods of millions of people.

## 1. Introduction

The grasslands of Madagascar have long been considered degraded wastelands (e.g. [1–3]). Consequently, little effort has been made to investigate their ecology, yet these grasslands cover over half the island [4]. Recently, endemic grass lineages have been found to have evolved in Madagascar many millions of years before human arrival [5,6]. It has been suggested that modern grasslands expanded significantly via people introducing cattle and bringing fire [7]. Secondary grassy ecosystems, the result of forest degradation and agricultural conversion do exist across the island [8] but their distinction from ancient grasslands remains confusing. However, modern fire regimes in Malagasy grasslands have been identified where humans have limited influence, with fire return intervals of one to three years [9], similar to fire regimes of African grasslands with

similar climates and where grasslands are considered ancient [10–14]. Humans arrived around 10 500 BP and anthropogenic landscape modification *ca.* 2300 BP led to megafaunal extinction alongside the introduction of cattle, where both overlapped by around 1500 years [15–17]. Despite being of fundamental relevance to supporting livelihoods, conservation and resolving contentions over ancient Malagasy ecosystems (e.g. [18,19]), there has been sparse examination of the ecology of grasses (e.g. [20]).

In the past, a diverse vertebrate herbivore assemblage of now-extinct primates, hippopotamuses, elephant birds and giant tortoises inhabited the island [21] which were suggested to have used grasslands [7]. Hippopotamuses and giant tortoises are prime grazer candidates [18,19] but carbon isotope data exist for only a few specimens from the grassy centre of the island and evidence to support a grazer assemblage is limited [19]. Existing isotopic data show that hippopotamuses and tortoises consumed primarily $C_3$ plants with a variable $C_4$ plant component [19], although emerging evidence supports a more mixed $C_3$–$C_4$ diet [22]. Understanding links between grasslands and the extinct fauna is crucial to determining the pre-settlement extent of the $C_4$-dominated grassy biomes.

Tropical grasslands the world over are structured by fire and grazing interacting with climate and soils [14,23]. As top-down controls, fire and grazing transform organic materials to modify community structure and act as evolutionary agents [24]. However, each process has different requirements. Grazing mammals require nutritious nitrogen-rich moist forage while fire consumes senesced carbon-rich plant material [25]. Thus, frequent fire versus frequent grazing leads to divergences in community composition [25,26]. Fire-associated grasses have traits promoting flammability and fire tolerance, while grazing lawn grasses have functional traits enabling proliferation under intense grazing but only where grazing is regular and concentrated. That is, the competitiveness and tolerance of grass life-history strategies to each consumer control initiates positive feedbacks between plant functional traits and consumer controls [25].

The main argument for the anthropogenic assembly of Malagasy grasslands is low diversity [1,3] and a lack of geographical structure [27]. However, the diversity of the Malagasy grass flora is in line with most other islands of a similar size while endemicity is higher, at approximately 40% [5], and the geography of Malagasy grasslands has been little investigated [28]. Given that similar expanses of grasslands occur in a similar range of rainfall across Africa, Australia and the Americas where grasslands are recognized as natural and ancient [10–14], it is puzzling Madagascar should be an anomaly in biome distributions. On the African continent, compositional differentiation among grasslands can be linked to grazing and fire regimes that promote functionally divergent grassy ecosystems (e.g. [25,26,29]). Here, we develop an overdue new understanding of the functional ecology and biogeography of grasslands across central Madagascar.

## 2. Material and methods

### (a) Study sites
We sampled the grass community at 71 sites across the central ecoregion of Madagascar among the regions of Ibity, Itremo, Isalo, Ankazobe and Antsirabe ([30]; electronic supplementary

material, figure S1). Data from 21 sites were from Solofondranohatra *et al.* [31]. The vegetation across the central ecoregion is predominantly extensive grassland and savannah woodland with some closed forest [4]. Mean annual rainfall ranges between 1200 mm and 1700 mm (Worldclim Global Climate Data version 2; [32]; see the electronic supplementary material, figure S2) with a five- to seven-month dry season [33]. Soils are primarily ferralitic on sandstone and basement gneiss [4].

### (b) Data collection
#### (i) Grass species community composition
Grass species sampled at one site define a community in our analyses. In the field, community composition was quantified using the sampling method described in Vorontsova *et al.* [5], to capture grass species diversity and relative frequency in a uniform vegetation area with a minimum area of 60 m × 60 m. All grass species within a centre circle plot of 1 m diameter were recorded and, from this centre point, four 25 m transects, each following a random direction (based on a compass bearing) from the point of origin were laid out. Along each transect, circular plots of 1 m diameter were sampled at 5 m intervals, representing grass species composition over 16.5 m². Species lists and their occurrences are presented in the electronic supplementary material, table S1.

#### (ii) Species rarity
Species were defined as rare based on two criteria: (i) the maximum frequency of a species within a community was less than five of 21 circular plots, and (ii) the species occurred in five or fewer of the 71 grass communities assessed. Analyses involving grass functional traits were undertaken on species that were not rare. Based on this assessment, grass functional traits of 41 common grass species were collected. While a further 26 species were recorded, their functional traits were not assessed owing to rarity.

#### (iii) Grass functional traits related to fire and grazing
Functional traits capture dimensions of life-history strategies via quantifying morphology and architecture. We measured five grass functional traits related to flammability, palatability and tolerance to fire and grazing: (i) plant height, defined as leaf table height (the height measured and visually estimated to correspond to the *ca.* 80th quantile of leaf biomass), has consequences for light competition with taller grasses effective at competing for light [34], and flammability as taller grasses are generally high in biomass [35]; (ii) leaf thickness influences palatability with thick tough leaves being less digestible [36], and flammability as leaves with higher $C:N$ ratios are more flammable; (iii) ratio of leaf width to leaf length reflects leaf shape with wide short leaves preferred by grazers as palatable and long narrow leaves igniting easily and burning intensely [37]; (iv) bulk density defined as mass per unit volume, relates to palatability and flammability. High bulk density grasses provide more forage per bite whereas low bulk density grasses provide aerated fuel beds [25]; and (v) architectural growth form of a grass can define the location of meristematic tissues to resist grazing and fire [38]. Full details on functional traits and collection methods are provided in the electronic supplementary material, table S2.

#### (iv) Environmental variables
Environmental data for Madagascar is of poor quality with few reliable weather stations, necessitating the use of global and modelled products. We calculated four environmental variables to examine the geography of grass communities: (i) mean annual precipitation (MAP) was obtained from Worldclim

Global Climate Data [32] as proxy for productivity [39]; (ii) per cent sand in the top 10 cm of soil (sand per cent) was obtained from Harmonized World Soils Database [40] that reflects soil water holding capacity where sandy soils have low water holding capacity, thus partly capturing patterns of landscape water availability; (iii) the presence/ absence of fire was scored for each site based on interviews with local communities and land managers; and (iv) distance to road was a proxy for grazing pressure and quantified using the national roads layer for Madagascar [41] with three levels of road (tarred, untarred and track). Cattle are the dominant grazers across Madagascar, associated with human communities that are generally close to roads. Some main roads through the Central Highlands also follow river valleys and can also reflect landscape water availability and soil properties that is also important to shaping potential cattle densities. Values of these environmental variables across our 71 studied sites are given in the electronic supplementary material, figure S2.

## (c) Analyses

### (i) Modelling grass species assemblages

Generalized latent variable models were used to determine whether distinct grass assemblages could be identified across sites based on the patterns of species co-occurrences [42] across 71 communities. Rare species as defined above were omitted from the analysis because they typically contribute little interpretive value while adding noise to the statistical solution [43]. Accordingly, 41 of 67 species were used in our assemblage analyses.

Relative species frequencies of each species in each community was the response variable. Candidate models comprised the full set of additive permutations of four environmental variables in addition to a single unobserved predictor (latent variable). All environmental variables were scaled prior to analysis, with MAP and distance to road being base-10 log transformed to meet model assumptions. Models were fitted in R (R version 3.0.2 [44]; using the gllvm package [45]), specifying a negative binomial error distribution and accounting for spatial autocorrelation by including site latitude and longitude as variables.

### (ii) Identifying grass species assemblages and environmental associations

Model comparisons were based on the Akaike information criterion (AIC; [46,47]). Using the most supported model, species assemblages were identified based on the matrix of residual correlations along with histograms of residual correlations for each species to identify natural breaks in residual correlation values (electronic supplementary material, figure S3). Residual correlation values range from −1 to +1. Based on the histograms, species grouped naturally into two assemblages where values were (i) greater than 0.1 and (ii) less than or equal to 0.1. Species with residual correlations ranging from −0.1 to +0.1 represent a lack of any association and species were not classified into either assemblage as they may be equally likely and unlikely to co-occur.

Rare species not incorporated into the gllvm analyses were assigned a post-hoc assemblage group, made possible by the very strong species co-occurrence patterns. To classify these 21 species, each community was assigned an assemblage group based on the dominant proportion of species in each assemblage group. Assemblage assignments for the 21 rare species enabled us to undertake analyses of phylogenetic conservatism described later. Finally, the relationship between each environmental correlate and species assemblage was assessed by plotting model coefficients of environmental correlate values for each assemblage group using boxplots.

### (iii) Identifying grass functional types

We sought to identify syndromes of functional traits that represent functionally similar species. These functional groups could then be cross-referenced with assemblage groups. Functionally similar species were identified using hierarchical clustering on principal components of the five functional traits described above for the 41 common grass species. Clustering used the Ward method based on Euclidian distance. The final number of clusters was determined using the sum of within-cluster inertia [48] where the final number of clusters corresponded with the highest relative loss of inertia. Functional trait values were then plotted for each cluster using violin plots and clusters were compared using analysis of variance (ANOVA).

### (iv) Species evolutionary history

To explore phylogenetic patterns of grass species relative to assemblage groups and functional traits, we extracted the Bayesian time-calibrated phylogenetic tree of the species from a large analysis of Malagasy grasses [6]. *Digitaria thouaresiana*, *Eragrostis atrovirens* and *Schizachyrium exile* had no DNA sequences available and were not included. *Paspalum scrobiculatum* was replaced by the only species within the Paspaleae tribe (*Hildaea pallens*) in Hackel *et al.* [6], and *Axonopus compressus* was inserted based on its estimated divergence from *Paspalum* in Christin *et al.* [49].

Three species level attributes were plotted against the phylogenetic tree of 64 species, these were: (i) assemblage group; (ii) functional group; and (iii) endemicity (obtained from the [50]).

Four analyses were then undertaken to test: (i) differences in species richness [51] and phylogenetic diversity [52] between the two assemblage groups; (ii) differences in endemicity between the two assemblage groups; (iii) distribution of species functional traits along the phylogeny between the two assemblage groups; and (iv) phylogenetic conservatism of functional traits. Each test respectively used: (i) a generalized linear model with a Poisson distribution and log link function; (ii) a two-proportions $z$-test; (iii) a phylogenetic ANOVA using 'phytools' package [53]; and (iv) an estimation of Blomberg's K [54] with the 'phylosig' function using 999 numbers of tree shuffling randomization.

## 3. Results

## (a) Assemblage groups

Residual correlations very clearly identified two species groups (figure 1). The most supported model generating these groups included MAP, distance to road and the presence/absence of fire as environmental correlates (AIC = 4904.07, ΔAIC to second-best model = 2.18, figure 1; electronic supplementary material, table S3). Group 1 (top of the correlation matrix) was composed of species highly likely to co-occur with significant positive correlations (figure 1). Species from group 1 were highly unlikely to co-occur with any species in group 2, all of which are characterized by significant negative correlations (figure 1). Six species had residual correlation values ranging from −0.1 to +0.1 (figure 1; electronic supplementary material, figure S3) and were not classified into either assemblage. Assemblage groups corresponding to each analysed species are presented in the electronic supplementary material, table S1.

## (b) Linking assemblage groups with environment

MAP and the presence of fire had largely negative associations with assemblage group 1, and positive associations with assemblage group 2 (figure 2). Two species had very large coefficients related to rainfall. These were *Brachiaria*

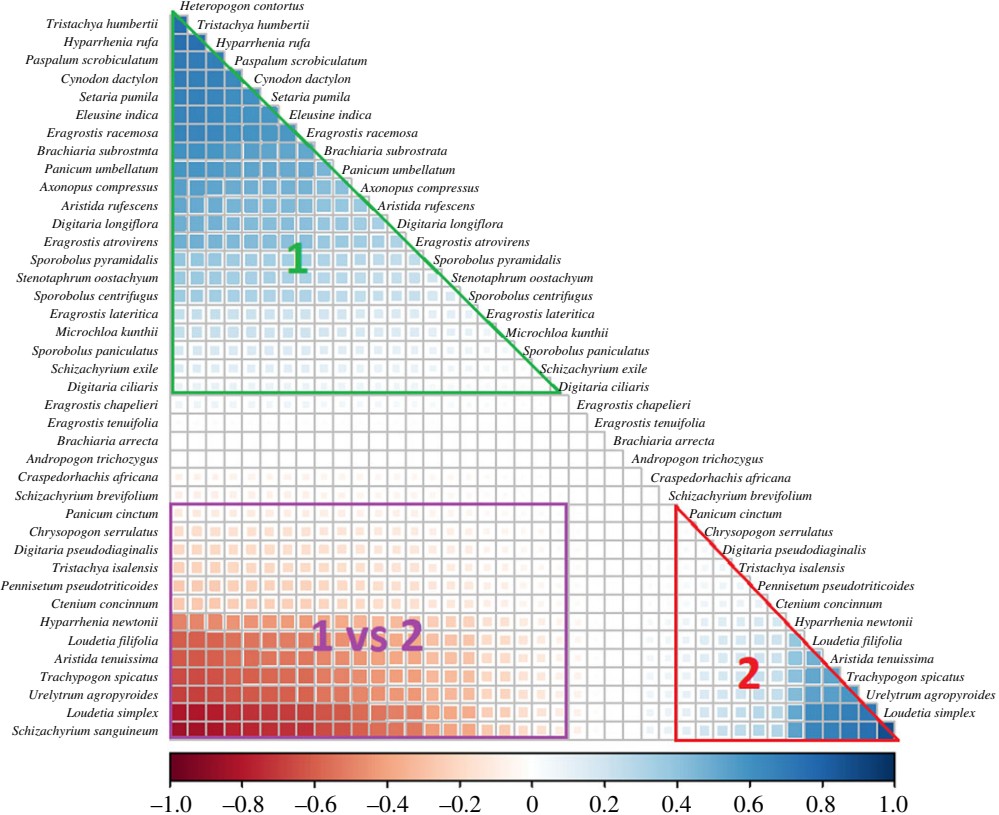

**Figure 1.** Residual correlation values between pairs of 41 grass species based on 71 grassland communities. Values indicate the likelihood of pairwise species co-occurrence that identified two major grassland assemblages: 'group 1' (top of the matrix) and 'group 2' (bottom right of the matrix). group 1 species are highly likely to co-occur but not with species in group 2. Significant ($p < 0.05$) positive correlations are represented by blue cells, and significant negative associations correspond to red cells. Non-significant associations are blank. Correlation values are estimated from a generalized linear latent variable model incorporating mean annual precipitation, the presence/absence of fire, distance to road and a single latent variable. (Online version in colour.)

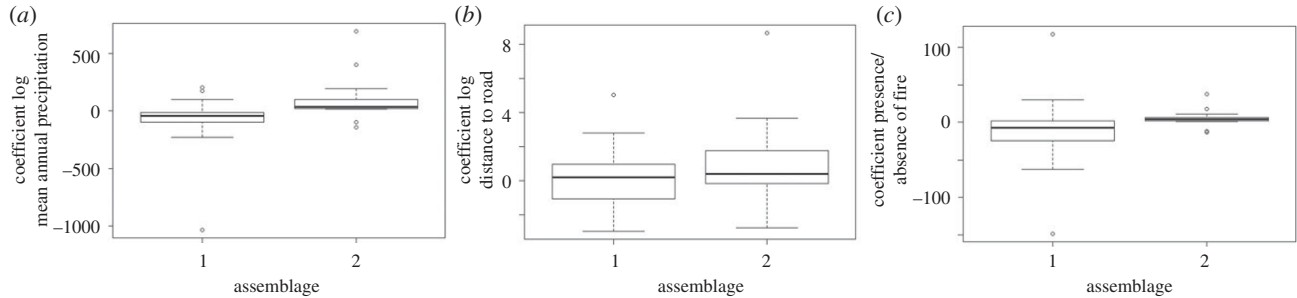

**Figure 2.** Model coefficients of environmental correlates compared between grass assemblage groups. Coefficients are related to (a) mean annual precipitation (mm yr$^{-1}$), (b) distance to road (m) and (c) the presence or absence of fire.

*subrostrata* and *Pennisetum pseudotriticoides* with coefficients, respectively, of −1030 and 690. By contrast, *Brachiaria subrostrata* had a strongly negative coefficient related to fire presence (−148). Extreme coefficients relate to the absence of these species from many communities with the model for mean frequency appropriately fitted on a log scale. Distance to road has variable relationship with assemblage 1 and mainly positive relationships with assemblage 2 (figure 2).

## (c) Syndromes of grass functional traits

Hierarchical clustering identified three functional groups of species associated with grazing and fire alongside an intermediate group (harbouring traits between the two groups)

(figure 3a). Significant differences were found between all numerical mean trait values of the three groups ($p < 0.001$, figure 3b). The grazing group of 14 species, more than half of which are mat forming (57.1% of the group) and with all sampled mat-forming species within this group are short grasses with high bulk densities, and short wide thin leaves. Leaf width to length ratio and bulk density were similar between grazing and intermediate groups (all $p > 0.05$) but far higher than the fire group (all $p < 0.001$). The fire group comprises 23 species, all of which are tall caespitose grasses with thicker leaves, low bulk density and low leaf width to length ratios compared to the grazing group (all $p < 0.001$). Species in the intermediate group have similar bulk densities to species in the fire group ($p > 0.05$).

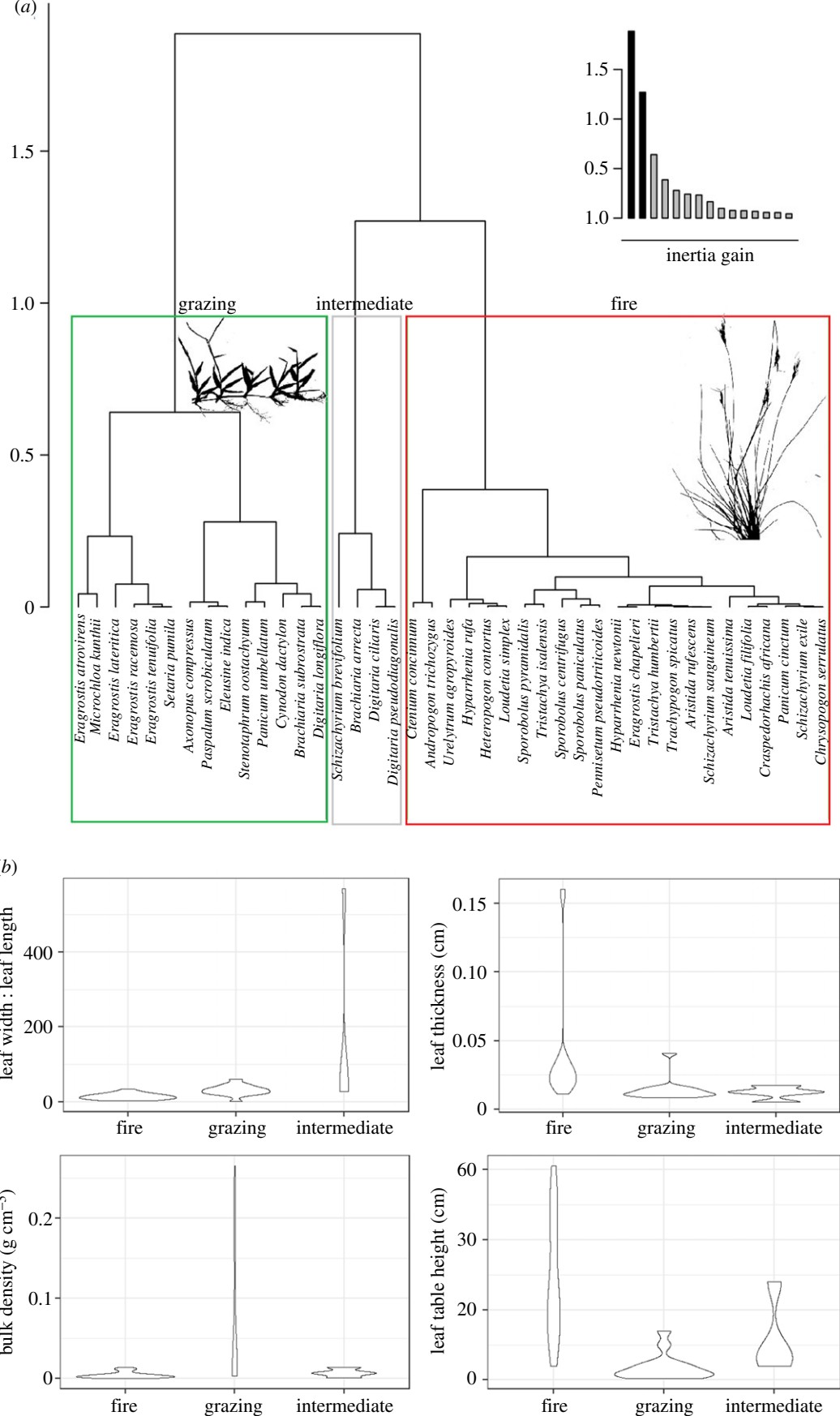

**Figure 3.** Three clusters of grass species representing significant differences among groups in three functional traits. (*a*) Dendrogram produced via hierarchical clustering on principal component (HCPC) of 41 grass species based on growth form, leaf width to length ratio, leaf thickness, bulk density and leaf table height. Three functional groups are supported and interpreted as related to: grazing; intermediate (traits enabling tolerance of some level of both grazing and fire) and fire. Black silhouettes represent typical grazing (*Paspalum conjugatum*) and fire (*Loudetia filifolia*) grass morphologies. (*b*) Violin plots of four functional traits per functional group from the HCPC dendrogram. There are significant differences in all the traits between the functional clusters ($p < 0.001$ for each). (Online version in colour.)

**Table 1.** Number of grass species in assemblage and functional groups. (Number of endemics per group are given in brackets.)

| | assemblage group 1 (grazing) | assemblage group 2 (fire) | total species per functional group (including species which were not part of either assemblage groups) |
|---|---|---|---|
| grazing group | 13 | 0 | 14 (4 endemics) |
| intermediate group | 1 | 1 | 4 (0 endemic) |
| fire group | 8 | 12 | 23 (7 endemics) |
| total per assemblage group | 22 (5 endemics) | 13 (5 endemics) | |

## (d) Linking assemblage and functional groups

We found high correspondence between the assemblage and functional analyses (table 1). Thirteen of 14 species in the grazing functional group (92.85%) are found in assemblage 1. Of the 22 species within assemblage 1 (59.1%) were clustered in grazing group. By contrast, assemblage 2 is strongly associated with the fire functional group with 12 of the 13 species in assemblage 2 found in the fire grass functional group. Chi-square test result showed that functional and residual groups have a significant relationship ($p = 0.001$). Among the 41 species for which there are functional data, there are 11 endemic species, of which five each are, respectively, found in assemblages 1 and 2. Four endemic species are found in the grazing-adapted functional group, seven in the fire-adapted functional group and none in the intermediate group. Based on the evidence, assemblage 1 represents a suite of grazer-maintained communities while assemblage 2 represents a suite of fire-maintained communities.

## (e) Species evolutionary history

The two assemblages are phylogenetically over-dispersed (electronic supplementary material, figure S4). Of the 67 sampled species, 31% are endemic. Twelve endemic species are associated with the fire-maintained assemblage and eight with the grazing-maintained assemblage (figure 4). One endemic species (*Andropogon trichozygus*) has residual correlation values ranging from −0.1 to +0.1 and is among the species not classified into either assemblage. There are no significant differences between the proportion of endemics in the two assemblages ($p > 0.05$) while accounting for phylogeny. However, a phylogenetic ANOVA found that variances within assemblages are associated with grass leaf table height ($p = 0.008$, $F = 4.26$) and bulk density ($p = 0.04$, $F = 2.59$) but not leaf size or thickness. The species richness is similar between the two assemblage groups, and phylogenetic diversity within the grazing-maintained assemblage is significantly higher than the fire-maintained assemblage (electronic supplementary material, figure S4). No significant phylogenetic signal was found for any of the functional traits, indicating that these are evolutionarily labile (all $p > 0.05$ for the four numerical traits).

## 4. Discussion

In Madagascar, grasslands are far from a homogenous landscape but, much like in continental Africa, are shaped by the contrasting processes of fire and grazing that promote differentiation in community composition where constituent species have diverging syndromes of functional traits. In our research, Malagasy grass assemblages shaped by grazing

and fire each have approximately 30–40% endemism (table 1, figure 4). These endemic grazer and fire specific species pre-date human arrival (*ca*. 10 500 B.P., [16,17]) by millions of years, with a divergence age range of 1–7 million years [6], suggesting that grazing animals and fire shaped community assembly in a functionally comparable way to grassland ecosystems in Africa well before human arrival.

The Malagasy grazing lawn assemblage (assemblage 1 and grazing functional group; figures 1, 3 and 5) is characterized by short, mat-forming, high bulk density grasses with short wide thin leaves. Grazing lawns can only spread and persist under consistent concentrated grazing that limits light competition from other grass species [55,56] but also requires that grass species keep meristematic tissue at or below the soil surface, and thus inaccessible to grazers, to tolerate such consistent grazing. The fire grass assemblage (assemblage 2 and fire functional group; figures 1, 3 and 5) is characterized by similar species richness and lower phylogenetic diversity relative to the grazing lawn assemblage (electronic supplementary material, figure S4) with tall caespitose grasses with low bulk density and longer, narrower and thicker leaves. Tall grasses, usually with a high aboveground biomass quantity and low bulk density (i.e. sparse architecture), are highly flammable and promote fire [35]. The presence of endemic fire grass species strengthens the evidence that some extent of fire-maintained grasslands is an ancient part of the region.

Despite the congruence identified between assemblages and functional groups, a small suite of species did not match between analyses. We interpret these species as being potentially able to persist in communities shaped either by fire or grazing through tolerating both consumers to some degree. These species, such as *Hyparrhenia rufa*, *Heteropogon contortus* and *Sporobolus pyramidalis*, also have pan-African or even cosmopolitan range sizes as would be expected if a species can tolerate a wide range of disturbance conditions [57]. In our dataset, these species were functionally clustered within the fire grasses, but possibly as a product of traits being sampled where species were first encountered in our surveys, i.e. in frequently burnt communities, while these species were also found elsewhere.

Madagascar's extinct megafauna, including hippopotamuses, giant tortoises, elephant birds and giant lemurs survived well into the Holocene [7,58], and their extirpation *ca*. 1200 BP was well after anthropogenic landscape modification is noted in the palaeo-record [16,17]. Malagasy hippopotamuses, members of the derived genus *Hippopotamus* arrived in Madagascar in the Quaternary [59,60]. A recent isotope record suggest that hippopotamuses in central Madagascar consumed a mixed diet of $C_3$ and $C_4$ plants in an open ecosystem [22] although previous isotope data suggested a primarily $C_3$ diet where the majority of grasses

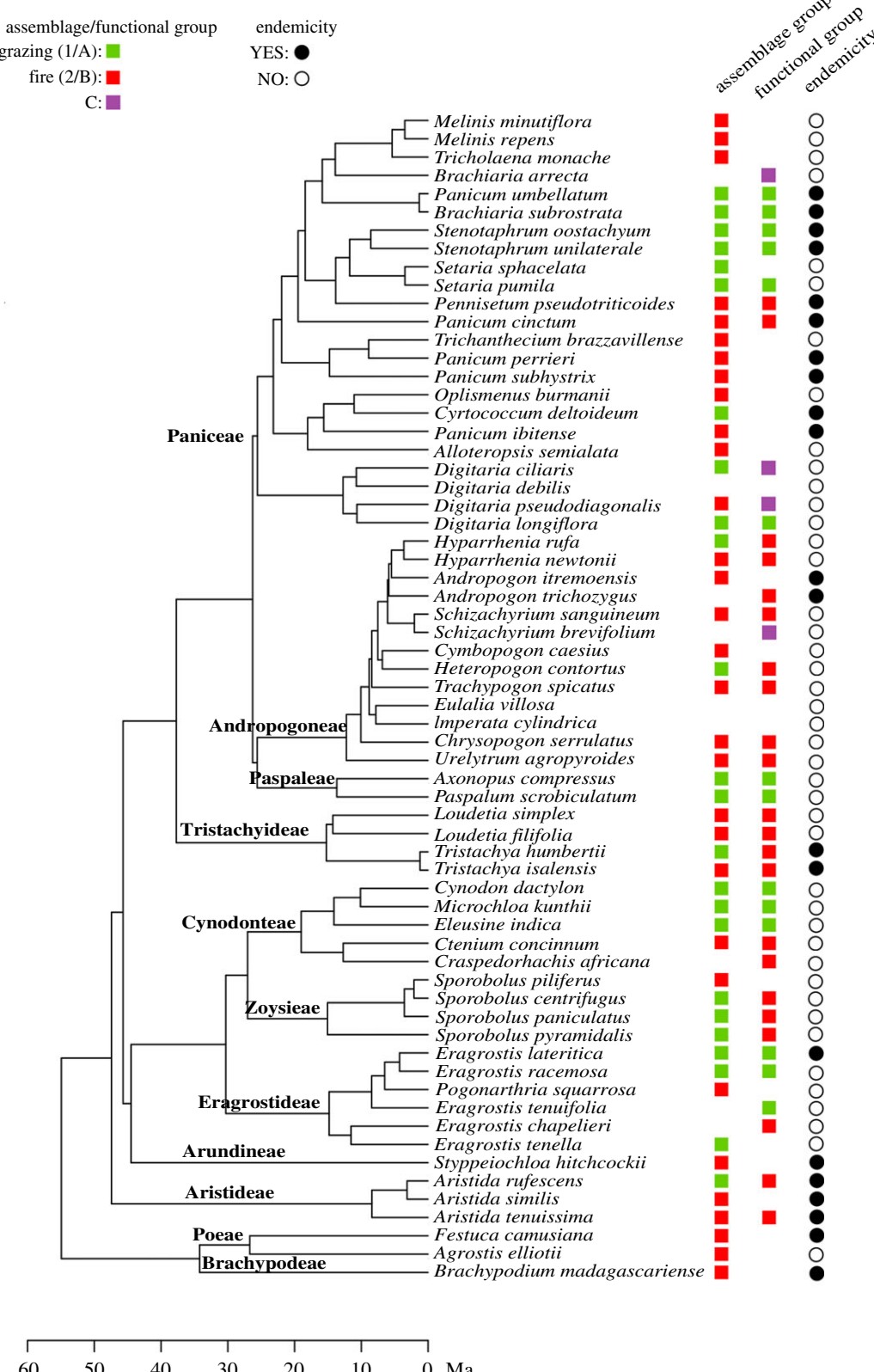

**Figure 4.** Phylogenetic tree of all 64 grass species mapped to: two assemblage groups (for all species except those that could not be attributed to either assemblage); three functional groups of the 39 common species (*Eragrostis atrovirens* and *Schizachyrium exile* are not included owing to lack of available sequences); and species endemicity. Functional group C corresponds to the intermediate group in figure 3. (Online version in colour.)

in the Central Highlands are $C_4$ [19]. In Africa, hippopotamuses are short grass grazing specialists that play a crucial role in initiating and maintaining grazing lawns in areas of high rainfall [56,61] similar in rainfall to our study sites. Although hippopotamuses isotopic values in Africa are higher ([62] ($\delta^{13}C = -3.6‰$), [63] ($\delta^{13}C = -3.5‰$)) compared to Malagasy hippopotamuses ([22] ($\delta^{13}C = -15.9‰$)), it does

suggest some level of a mixed $C_3$ and $C_4$ diet. Samonds *et al.* [22] suggest that Malagasy hippopotamuses may be ecologically comparable to the African pygmy hippopotamus, *Choeropsis liberiensis*. A mixed diet would also be supported by the abundance of $C_3$ forbs common to grazing lawns that can be highly palatable [64]. In Madagascar, tortoises were also known to consume some proportion of $C_4$ and/

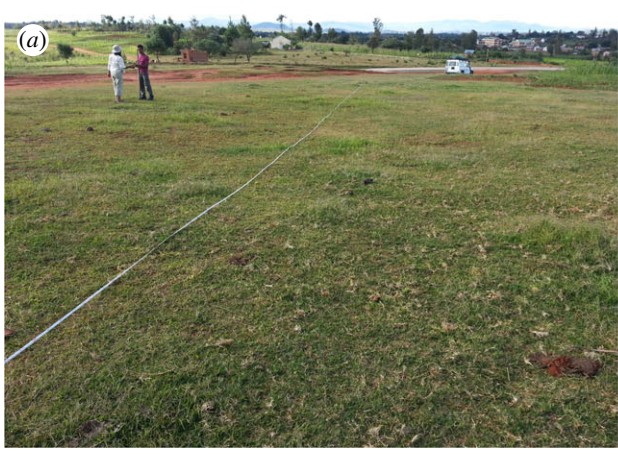
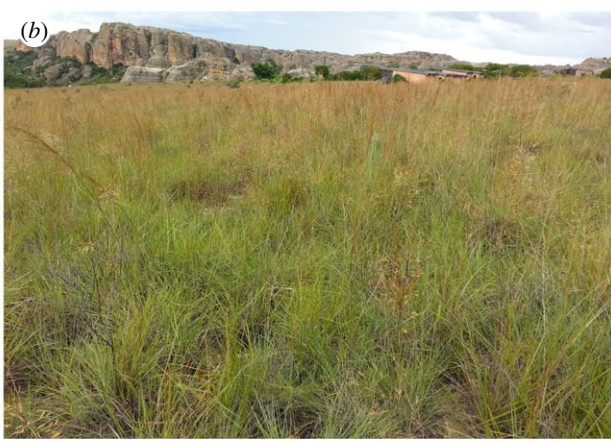

**Figure 5.** Examples of typical grasslands in the Madagascar Central Highlands: (*a*) a grazing lawn in Ibity, containing 18 species and dominated by *Cynodon dactylon* (NE), *Panicum umbellatum* (E) and *Digitaria longiflora* (NE); and (*b*) fire-maintained grassland in Isalo containing eight species and dominated by *Loudetia simplex* (NE) and *Loudetia filifolia* (NE). NE = not endemic, a grass species with a distribution that spans Africa and Madagascar. E = endemic, a grass species restricted to Madagascar and Mascarene Islands. (Online version in colour.)

or CAM plant material [19,65] and both $C_3$ and $C_4$ plants on the Mascarene islands [66]. A high density of tortoises can effectively keep grass short and unable to carry fire (e.g. [67]). It should be noted that isotope records in Madagascar are not complete in the Central Highlands possibly as preservational environments are limited and areas of possible preservation have long been suited to agriculture. We suggest the ecology of the grasses examined here demonstrates that in the early Pliocene megagrazers, most likely hippopotamuses and giant tortoises, were instrumental in the evolution and assembly of the Malagasy Central Highlands obligate grazing lawn flora (figure 4), and that cross-disciplinary efforts to reconcile palaeo and ecological data are much needed.

The geography of grazing lawns and fire grasslands is not random but related to rainfall, distance to roads and the presence of fire (figure 2) that also represent a legacy of human colonization and patterns of modern land use. Sites with higher rainfall were more likely to experience fire, while sites with lower rainfall were more likely associated with grazing. Across the rainfall gradient, sites located near roads are more likely subject to intensive concentrated grazing. Undoubtedly, the modern dynamics of grazing lawns in Madagascar are shaped by cattle raised close to roads (or waterways), where people live and can manage them relatively easily in terms of forage and safety. However, the associations of species dependent upon grazing are probably ancient, evidenced by the species diversity and endemicity. Cattle, hippopotamuses and grazing tortoises share key functional similarities, they prefer highly palatable grasses with high bulk density to maximize intake of nutritious food per bite. McCauley *et al.* [68] showed that a mixture of herbivores (including cattle and hippopotamuses) and removal of hippopotamuses on grazing lawns in East Africa similarly impacted grassland diversity and structure, suggesting some functional equivalence between hippopotamuses and livestock. The replacement of one grazer with another is unlikely to have substantially reshaped diversity where an obligate grazing flora already existed. While grazing lawns in Africa are maintained by a diversity of wild mammal grazers, cattle increasingly maintain grazing lawns owing to the vast and extensive displacement of native grazers with livestock. In Africa, grazing lawns also support a diversity of grass species [56] with diversity in Malagasy grazing lawns

similar or greater [69,70]. The current decline and extinction of African megafauna may well be an analogy of the historic megafaunal extinctions in Madagascar, where productive landscapes now used for cattle rearing are fundamentally underpinned by an ancient obligate grazing adapted flora, a product of millions of years of grazer and grass coevolution.

Examination of the impacts of megafaunal extinction generally focuses on woody plants rather than grasses. While grasses can be long lived, it would be possible for grazing grasses in particular to be rapidly lost from ecosystems when over-topped by taller grasses or woody plants. Indeed, the temporal overlap between the megafaunal extinction and arrival of cattle may have been the salvation of the grazing adapted grass flora while also facilitating human colonization of the island. It will be crucial to understand the impacts of environmental change on these ancient grass assemblages with droughts increasing in frequency and severity. However, also much needed is identification of the limits of ancient and modern grassland ecosystems requiring collaboration across disciplines. In Madagascar, grasslands are dismissed as wastelands in need of forest restoration. Hence, grasslands are now the subject of extensive tree planting programmes, adopted as environmental policy for forest restoration, carbon sequestration and fuelwood production. The most commonly planted trees are exotic *Eucalyptus*, *Acacia* and *Pinus* species, species known as invasive elsewhere in the world. Food security in Madagascar is highly precarious and agriculture in the Central Highlands is dependent on abundant stream flow for rice production. If grasslands are an extensive ancient component of these Central Highlands landscapes, which is likely given the patterns of diversity, geography and endemism observed here, not only is planting of exotic tree species damaging, but at scale will probably reduce stream flow [71] with unforeseen environmental consequences in a changing climate. Malagasy grasslands require new science to help delimit pre-human versus modern limits linked to the assemblages identified here. There is a clear need for science to engage with regions hitherto dismissed as being of no value for the sake of future conservation, land management and livelihoods.

Data accessibility. Data available from the Dryad Digital Repository: https://dx.doi.org/10.5061/dryad.9ghx3ffd2 [72].

Authors' contributions. C.L.S., M.S.V. and C.E.R.L. designed research; C.L.S., M.S.V. and C.E.R.L. collected data; C.L.S., J.H. and G.P.H. analysed data; C.L.S. and C.E.R.L. wrote the paper; M.S.V., G.P.H., J.H., S.C. and J.V. contributed to the interpretation and the revision of the work.

Competing interests. We declare we have no competing interests.

Funding. This work was supported by the Ecologists in Africa grant (grant no. EA16/1046, 2016) from the British Ecological Society. G.P.H. acknowledges support from the National Research Foundation of South Africa (no. 114974, no. 115998). C.E.R.L. was supported by a GCRF International Collaboration Award from the Royal Society.

Acknowledgement. We thank William Bond for feedback on a manuscript draft and David Warton for statistical advice. Thanks to Madagascar National Parks (MNP), Direction générale des forêts (DGF) and Parc Botanique et Zoologique de Tsimbazaza (PBZT) for granting research permits. Thanks to Kew Madagascar Conservation Centre (KMCC) staff for support with permits and field work, as well as all local communities we worked with. Thanks to two anonymous reviewers for their insightful feedback.

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
