## [Reviewer comments · Proceedings of the Royal Society B: Biological Sciences]

Review History

RSPB-2019-2525.R0 (Original submission)

Review form: Reviewer 1

Recommendation

Accept with minor revision (please list in comments)

Scientific importance: Is the manuscript an original and important contribution to its field?

Good

General interest: Is the paper of sufficient general interest?

Good

Quality of the paper: Is the overall quality of the paper suitable?

Good

Is the length of the paper justified?

No

Should the paper be seen by a specialist statistical reviewer?

No

Do you have any concerns about statistical analyses in this paper? If so, please specify them explicitly in your report.

No

It is a condition of publication that authors make their supporting data, code and materials available - either as supplementary material or hosted in an external repository. Please rate, if applicable, the supporting data on the following criteria.

Is it accessible?

N/A

Is it clear?

N/A

Is it adequate?

N/A

Do you have any ethical concerns with this paper?

No

Comments to the Author

Review of Madagascar grasslands

This is an interesting paper reporting an analysis of grasses and grasslands in Madagascar. For nearly a century these grasslands were considered secondary systems derived from deforestation. This paper addresses the antiquity of the grasslands but, intriguingly, focuses on grasslands heavily impacted by grazers versus fire. Grazing lawns have been studied in Africa where they are maintained by specialist grazer mammals. The extinct grazers in Madagascar are assumed to have been hippos, giant tortoises, and, perhaps, some elephant birds. But isotopic evidence of the diet of sub-fossils is equivocal and has been interpreted as indicating lack of grass in the diets. How then, does one explain a whole suite of grass species, with many endemics, apparently adapted for heavy grazing pressure? This study is well designed and well implemented to show the distinct and divergent grass assemblages. There are clear differences in community assemblages and clear differences in the suite of functional traits of grass species. Species with short-spreading versus tall, caespitose architectures are sorted into short grass, heavily grazed versus tall grass frequently burnt assemblages. The phylogenetic analysis shows no deep phylogenetic conservatism so related taxa may occur in very different habitats. The authors make the interesting claim that the arrival of livestock 'rescued' the grazer-tolerant grasses from extirpation by more competitive tall grass species when the indigenous grazer guild was lost.

Though the study is impressive, the writing is quite turgid in places and probably overlong for Proc Roy Soc. The analytical methods are not standard so more care is needed in explaining simply and clearly what they show. The figures on my copy were largely illegible with very small fonts. The figure legends are uninformative as to what they represent and require a close reading of the manuscript to interpret. The paper deserves to be read by a wide readership, many of whom will not be botanically minded, so it is worth taking more effort over presentation.

Minor comments.

There are a number of minor spelling and grammatical errors;

1.15, understating = understanding; 1.78, arguments = argument; 1.140 tared= tarred; 1.226

Schizacyrium 1.230. Crasperorhachis; 1.316 Hemspoon; 1.346. Sentence beginning 'Given that' is not complete. L. 358. (Hempson et al 2015)? Replace ? with .

1.57-63. On diets of sub-fossils. The discovery of sub-fossils in the Central highlands by Samonds et al looks very promising for revealing more on diet of hippos. It is worth comparing their interpretation with that of isotopic studies of hippo diet in Africa such as Boisserie et al. 2005 Palaeogeography, Palaeoclimatology, Palaeoecology, 221(1-2), 153; and Cerling et al 2008, J.

Zoology. These show that hippos consumed a lot of C3 material with their C4 grasses. Indeed African hippo isotopic signatures seem close to that of Samonds et al. Based on your grass data, more effort is surely needed on exploring sub-fossil diets.

l.120-127. You need to explain, very briefly, why these 'functional' traits matter here. The reader should not have to go to the supplementary files for this essential information.

l. 296. Human arrival dates seem odd. Intro gives 6000 BP as date of arrival.

l.306. Is this the only reference on the need for heavy grazing to maintain short grass systems? Surely earlier papers are available to strengthen your argument.

l.339. I suggest reading Verweij et al. Oikos 2006 as a source of older literature on grazing lawns and a study where hippos maintain grazing lawns in high rainfall tall grass savannas.

l.366. 'grazing tortoises'. Grazing and browsing tortoises existed as inferred from presence or absence of a slot for raised neck.

l.379. Grasses can live for many centuries. However they can have rapid dynamic changes in dominance if conditions change.

Figures. The legends are obscure. Figure 1. Correlations of what with what? My (naïve?) expectation is that the blue cells should be +ve correlations and the red correlations -ve.

Figure 2. I could not make sense of this figure. Symbols were too small to read on x axes. What do the bars represent? Species? Plots? Figures should be easy to read without mining through the manuscript.

Figure 3. What is 'HCPC'? FGA, FGB, FGC are also very cryptic names for the three functional groups. Why not tell the reader 'grazer', 'fire' etc. in the legend or in the figure labels to aid interpretation of the data?

Figure 4. Quite a poor fit between community group classification and functional group classification. The problem seems to be greatest in the Sporobolus clade. Worth a comment?

Supplementary figure 7 is worth placing in the main text to indicate the two communities. It is remarkable to me that the grazing lawn has endemic grass species from Madagascar and not just weedy species imported from Africa.

Review form: Reviewer 2

Recommendation

Accept with minor revision (please list in comments)

Scientific importance: Is the manuscript an original and important contribution to its field?

Excellent

General interest: Is the paper of sufficient general interest?

Good

Quality of the paper: Is the overall quality of the paper suitable?

Excellent

Is the length of the paper justified?

Yes

Should the paper be seen by a specialist statistical reviewer?

No

Do you have any concerns about statistical analyses in this paper? If so, please specify them explicitly in your report.

No

It is a condition of publication that authors make their supporting data, code and materials available - either as supplementary material or hosted in an external repository. Please rate, if applicable, the supporting data on the following criteria.

Is it accessible?

No

Is it clear?

N/A

Is it adequate?

N/A

Do you have any ethical concerns with this paper?

No

Comments to the Author

This paper presents a coherent and well substantiated data set, convincingly analysed and presented, that weighs in strongly to support an ancient vs anthropogenic origin for Malagasy grasslands. The combination of field collected data, analysed using functional type data and from a trait perspective, together with the phylogenetic evidence presents a strong case indeed. I would prefer to see the work of Bond et al 2008 highlighted a little more in the introduction as a vanguard of stimulating and adding valuable fuel to this debate, but I note that he was consulted and is acknowledged for this.

In the discussion, these sentences are important for management implications, but not supported by a reference " Cattle, hippos and tortoises share key functional similarities, they prefer highly palatable 367 grasses with high bulk density to maximise intake of nutritious food per bite. The replacement 368 of one grazer with another is unlikely to have substantially reshaped diversity where an obligate 369 grazing flora already existed." A reference here is important. Further, are the authors implying that conservation of the grassland diversity is not under threat, and I wonder about current efforts towards afforestation for the purposes of charcoal production and carbon sequestration. This might be worth mentioning if the authors feel it is relevant.

Decision letter (RSPB-2019-2525.R0)

04-Dec-2019

Dear Ms Solofondranohatra,

I am writing to inform you that we have now received referees' reports on your manuscript RSPB-2019-2525 entitled "Fire and grazing determined grasslands of Madagascar are ancient assemblages".

Based on the advice of the Associate Editor and the referees, the manuscript has, in its current form, been rejected for publication in Proceedings B. We are all agreed that this is a potentially really valuable manuscript, and the analyses and the figures are excellent, but the writing of the paper needs substantial reworking. With this in mind we would like to invite you to resubmit provided the comments of the referees are fully addressed. However please note that this is not a provisional acceptance.

Sincerely,
 Professor Loeske Kruuk
 mailto: proceedingsb@royalsociety.org

Associate Editor
 Comments to Author:

This is a very interesting paper on the history of Madagascar's grasslands. Both reviewers found the topic interesting and the analysis informative and there is the opportunity to make a real contribution to the literature on this topic. The figures and the analysis are very cutting-edge. The writing about the analysis however, which is always a challenge with these complex models, could still be improved (see Reviewer 1's comments). In general I am confident that the reviewers comments will help greatly improve this manuscript.

A few other notes:

- The placement of some of the species on the phylogeny may (or may not) be solved via phyndr (depending on the topology) see Pennell et al. (2016).

- Figure 3 could definitely be improved with real units on the y-axis, and it's not clear how to interpret the log of a ratio.

- The lack of a phylogenetic linear model (PGLS) for an associations between functional group, community group, and endemism seems like an easy to fix omission.

Reviewer(s)' Comments to Author:
 Referee: 1
 Comments to the Author(s)
 Review of Madagascan grasslands

This is an interesting paper reporting an analysis of grasses and grasslands in Madagascar. For nearly a century these grasslands were considered secondary systems derived from deforestation. This paper addresses the antiquity of the grasslands but, intriguingly, focuses on grasslands heavily impacted by grazers versus fire. Grazing lawns have been studied in Africa where they are maintained by specialist grazer mammals. The extinct grazers in Madagascar are assumed to have been hippos, giant tortoises, and, perhaps, some elephant birds. But isotopic evidence of the diet of sub-fossils is equivocal and has been interpreted as indicating lack of grass in the diets. How then, does one explain a whole suite of grass species, with many endemics, apparently adapted for heavy grazing pressure? This study is well designed and well implemented to show the distinct and divergent grass assemblages. There are clear differences in community assemblages and clear differences in the suite of functional traits of grass species. Species with short-spreading versus tall, caespitose architectures are sorted into short grass, heavily grazed versus tall grass frequently burnt assemblages. The phylogenetic analysis shows no deep phylogenetic conservatism so related taxa may occur in very different habitats. The authors make the interesting claim that the arrival of livestock 'rescued' the grazer-tolerant grasses from extirpation by more competitive tall grass species when the indigenous grazer guild was lost.

Though the study is impressive, the writing is quite turgid in places and probably overlong for Proc Roy Soc. The analytical methods are not standard so more care is needed in explaining simply and clearly what they show. The figures on my copy were largely illegible with very small fonts. The figure legends are uninformative as to what they represent and require a close reading of the manuscript to interpret. The paper deserves to be read by a wide readership, many of whom will not be botanically minded, so it is worth taking more effort over presentation.

Minor comments.

There are a number of minor spelling and grammatical errors;

l.15, understating = understanding; l.78, arguments = argument; l.140 tared= tarred; l.226 Schizacyrium l.230. Crasperorhachis; l.316 Hemspoon; l.346. Sentence beginning 'Given that' is not complete. L. 358. (Hempson et al 2015)? Replace ? with .

l.57-63. On diets of sub-fossils. The discovery of sub-fossils in the Central highlands by Samonds et al looks very promising for revealing more on diet of hippos. It is worth comparing their interpretation with that of isotopic studies of hippo diet in Africa such as Boisserie et al. 2005 *Palaeogeography, Palaeoclimatology, Palaeoecology*, 221(1-2), 153; and Cerling et al 2008, *J. Zoology*. These show that hippos consumed a lot of C3 material with their C4 grasses. Indeed African hippo isotopic signatures seem close to that of Samonds et al. Based on your grass data, more effort is surely needed on exploring sub-fossil diets.

l.120-127. You need to explain, very briefly, why these 'functional' traits matter here. The reader should not have to go to the supplementary files for this essential information.

l. 296. Human arrival dates seem odd. Intro gives 6000 BP as date of arrival.

l.306. Is this the only reference on the need for heavy grazing to maintain short grass systems? Surely earlier papers are available to strengthen your argument.

l.339. I suggest reading Verweij et al. *Oikos* 2006 as a source of older literature on grazing lawns and a study where hippos maintain grazing lawns in high rainfall tall grass savannas.

l.366. 'grazing tortoises'. Grazing and browsing tortoises existed as inferred from presence or absence of a slot for raised neck.

l.379. Grasses can live for many centuries. However they can have rapid dynamic changes in dominance if conditions change.

Figures. The legends are obscure. Figure 1. Correlations of what with what? My (naïve?) expectation is that the blue cells should be +ve correlations and the red correlations -ve.

Figure 2. I could not make sense of this figure. Symbols were too small to read on x axes. What do the bars represent? Species? Plots? Figures should be easy to read without mining through the manuscript.

Figure 3. What is 'HCPC'? FGA, FGB, FGC are also very cryptic names for the three functional groups. Why not tell the reader 'grazer', 'fire' etc. in the legend or in the figure labels to aid interpretation of the data?

Figure 4. Quite a poor fit between community group classification and functional group classification. The problem seems to be greatest in the Sporobolus clade. Worth a comment?

Supplementary figure 7 is worth placing in the main text to indicate the two communities. It is remarkable to me that the grazing lawn has endemic grass species from Madagascar and not just weedy species imported from Africa.

Referee: 2

Comments to the Author(s)

This paper presents a coherent and well substantiated data set, convincingly analysed and presented, that weighs in strongly to support an ancient vs anthropogenic origin for Malagasy grasslands. The combination of field collected data, analysed using functional type data and from a trait perspective, together with the phylogenetic evidence presents a strong case indeed. I would prefer to see the work of Bond et al 2008 highlighted a little more in the introduction as a vanguard of stimulating and adding valuable fuel to this debate, but I note that he was consulted and is acknowledged for this.

In the discussion, these sentences are important for management implications, but not supported by a reference " Cattle, hippos and tortoises share key functional similarities, they prefer highly palatable 367 grasses with high bulk density to maximise intake of nutritious food per bite. The replacement 368 of one grazer with another is unlikely to have substantially reshaped diversity where an obligate 369 grazing flora already existed." A reference here is important. Further, are the authors implying that conservation of the grassland diversity is not under threat, and I wonder about current efforts towards afforestation for the purposes of charcoal production and carbon sequestration. This might be worth mentioning if the authors feel it is relevant.

Author's Response to Decision Letter for (RSPB-2019-2525.R0)

See Appendix A.

RSPB-2020-0598.R0

Review form: Reviewer 1

Recommendation

Accept with minor revision (please list in comments)

Scientific importance: Is the manuscript an original and important contribution to its field?

Good

General interest: Is the paper of sufficient general interest?

Good

Quality of the paper: Is the overall quality of the paper suitable?

Good

Is the length of the paper justified?

Yes

Should the paper be seen by a specialist statistical reviewer?

No

Do you have any concerns about statistical analyses in this paper? If so, please specify them explicitly in your report.

No

It is a condition of publication that authors make their supporting data, code and materials available - either as supplementary material or hosted in an external repository. Please rate, if applicable, the supporting data on the following criteria.

Is it accessible?

Yes

Is it clear?

Yes

Is it adequate?

Yes

Do you have any ethical concerns with this paper?

No

Comments to the Author

Thanks for the extensive edits and the helpful explanations of responses. The paper reads far better, the figures are easily interpreted now with the revised figure legends, and the edits to the discussion enhance the importance of the work. Unfortunately quite a few minor edits have crept in and should be corrected in a very minor revision. Here is my list.

Abstract.

Line 25 to 28. Minor edit suggested.

Replace: . "Within each assemblage, levels of endemism, diversity and grass ages support these as ancient assemblages, and where grazed dependent grasses co-evolved with the now-extinct megafauna, likely hippos and giant tortoises."

With: Within each assemblage, levels of endemism, diversity and grass ages support these as ancient assemblages. Grazer-dependent grasses co-evolved with the now-extinct megafauna, likely hippos and giant tortoises.

1. 45. Lehmann et al., 2011

1. 46. Fix 'extinction alongside and the introduction of'

1.57. delete 'grazers'

1.59-62. Fix this sentence. E.g. replace 'data support' with 'data show that'

1.83. Well cited papers giving evidence for antiquity of grasslands (and especially savannas) include Cerling et al. 1997, Jacobs et al. 1999, Stromberg 2005, Edwards et al. 2010 along with Lehmann et al 2011.

- 1.97. why not replace this line with just 'closed forest'. Some are on drainage lines others are not, e.g. Ambohitantely etc.
- 1.107. Fix 'relative frequency of in a uniform..'
- 1.132-133. Something missing here. From the Discussion, it would seem to be 'palatable whereas long thin leaves ignite easily and burn intensely'.
- 1.270. Fix 'these evidence'
- 1.320. Delete 'a' at end of line
1. 326. 'data suggest'
- 1.334. fix 'a mixed of C3 and C4'. Incidentally, check journal usage for whether C4 and C4 have subscripts.
- 1.344 -345. Fix placing of commas to make sense of sentence.

Decision letter (RSPB-2020-0598.R0)

07-Apr-2020

Dear Ms Solofondranohatra,

I am pleased to inform you that your manuscript RSPB-2020-0598 entitled "Fire and grazing determined grasslands of central Madagascar represent ancient assemblages" has been accepted for publication in Proceedings B.

The referee and Associate Editor have recommended publication, but the referee has also suggests some minor revisions to your manuscript. Therefore, I invite you to respond to the referee's comments and revise your manuscript. Because the schedule for publication is very tight, it is a condition of publication that you submit the revised version of your manuscript within 7 days. If you do not think you will be able to meet this date please let us know.

- 1) A text file of the manuscript (doc, txt, rtf or tex), including the references, tables (including captions) and figure captions. Please remove any tracked changes from the text before submission. PDF files are not an accepted format for the "Main Document".
- 2) A separate electronic file of each figure (tiff, EPS or print-quality PDF preferred). The format should be produced directly from original creation package, or original software format. PowerPoint files are not accepted.
- 3) Electronic supplementary material: this should be contained in a separate file and where possible, all ESM should be combined into a single file. All supplementary materials

accompanying an accepted article will be treated as in their final form. They will be published alongside the paper on the journal website and posted on the online figshare repository. Files on figshare will be made available approximately one week before the accompanying article so that the supplementary material can be attributed a unique DOI.

If you wish to submit your data to Dryad (<http://datadryad.org/>) and have not already done so you can submit your data via this link [http://datadryad.org/submit?journalID=RSPB&manu=\(Document not available\)](http://datadryad.org/submit?journalID=RSPB&manu=(Document not available)) which will take you to your unique entry in the Dryad repository. If you have already submitted your data to dryad you can make any necessary revisions to your dataset by following the above link. Please see <https://royalsociety.org/journals/ethics-policies/data-sharing-mining/> for more details.

Yours sincerely,
Professor Loeske Kruuk
<mailto:proceedingsb@royalsociety.org>

Associate Editor
Board Member
Comments to Author:

The manuscript has improved greatly through the response to the reviewers' comments. There are, however, a number of minor issues that have appeared into the manuscript as identified by Reviewer 1. These will need attention in another revision.

Reviewer(s)' Comments to Author:

Referee: 1

Comments to the Author(s).

Thanks for the extensive edits and the helpful explanations of responses. The paper reads far better, the figures are easily interpreted now with the revised figure legends, and the edits to the discussion enhance the importance of the work. Unfortunately quite a few minor edits have crept in and should be corrected in a very minor revision. Here is my list.

Abstract.

Line 25 to 28. Minor edit suggested.

Replace: . "Within each assemblage, levels of endemism, diversity and grass ages support these as ancient assemblages, and where grazed dependent grasses co-evolved with the now-extinct megafauna, likely hippos and giant tortoises."

With: Within each assemblage, levels of endemism, diversity and grass ages support these as ancient assemblages. Grazer-dependent grasses co-evolved with the now-extinct megafauna, likely hippos and giant tortoises.

l. 45. Lehmann et al., 2011

l. 46. Fix 'extinction alongside and the introduction of'

l.57. delete 'grazers'

l.59-62. Fix this sentence. E.g. replace 'data support ' with 'data show that'

l.83. Well cited papers giving evidence for antiquity of grasslands (and especially savannas) include Cerling et al. 1997, Jacobs et al. 1999, Stromberg 2005, Edwards et al. 2010 along with Lehmann et al 2011.

l.97. why not replace this line with just 'closed forest'. Some are on drainage lines others are not, e.g. Ambohitantely etc.

l.107. Fix 'relative frequency of in a uniform..'

l.132-133. Something missing here. From the Discussion, it would seem to be 'palatable whereas long thin leaves ignite easily and burn intensely'.

l.270. Fix 'these evidence'

l.320. Delete 'a' at end of line

l. 326. 'data suggest'

l.334. fix 'a mixed of C3 and C4'. Incidentally, check journal usage for whether C4 and C4 have subscripts.

l.344 -345. Fix placing of commas to make sense of sentence.

Author's Response to Decision Letter for (RSPB-2020-0598.R0)

See Appendix B.

Decision letter (RSPB-2020-0598.R1)

15-Apr-2020

Dear Ms Solofondranohatra

I am pleased to inform you that your manuscript entitled "Fire and grazing determined grasslands of central Madagascar represent ancient assemblages" has been accepted for publication in Proceedings B.

Open Access

Paper charges

Sincerely,

Proceedings B

Appendix A

Associate Editor

Comments to Author:

This is a very interesting paper on the history of Madagascar's grasslands. Both reviewers found the topic interesting and the analysis informative and there is the opportunity to make a real contribution to the literature on this topic. The figures and the analysis are very cutting-edge. The writing about the analysis however, which is always a challenge with these complex models, could still be improved (see Reviewer 1's comments). In general I am confident that the reviewers comments will help greatly improve this manuscript.

Thank you for the positive feedback and helpful comments. In our re-submitted manuscript we hope to demonstrate that we have taken on board all comments to make the description of the methods and analytical results clear and succinct. The methods are now divided by sub-headings such that sections do not run into each other as previous, and within each section descriptions are clearly defined (lines 115-280).

Below we address each comment to improve the manuscript.

A few other notes:

- The placement of some of the species on the phylogeny may (or may not) be solved via phyndr (depending on the topology) see Pennell et al. (2016).

*Thank you for this comment. Indeed "phyndr" is a useful tool in replacing species that are not in the phylogeny with phylogenetically equivalent species. However, in our case with the missing three species (*Digitaria thouaresiana*, *Eragrostis atrovirens* and *Schizachyrium exile*), other species from the same genera are already included, and therefore we do not see that there is anything equivalent we could swap them with.*

- Figure 3 could definitely be improved with real units on the y-axis, and it's not clear how to interpret the log of a ratio.

Thank you. We amended Figure 3 to now depict non-transformed units on the y-axis relative to the three functional groups. We replaced boxplots with violin plots to better visualise trait distributions.

- The lack of a phylogenetic linear model (PGLS) for an associations between functional group, community group, and endemism seems like an easy to fix omission.

Thank you for pointing this out. Instead, of using a PGLS we added a phylogenetic ANOVA to test the distribution of species functional traits along the phylogeny and used the assemblage (previously called community) group as predictor variables. As the assignment of functional group was incomplete for all species, we only undertook this analysis for our assemblage groups.

Reviewer(s)' Comments to Author:

Referee: 1

Comments to the Author(s)

Review of Madagascan grasslands

This is an interesting paper reporting an analysis of grasses and grasslands in Madagascar. For nearly a century these grasslands were considered secondary systems derived from deforestation. This paper addresses the antiquity of the grasslands but, intriguingly, focuses on grasslands heavily impacted by grazers versus fire. Grazing lawns have been studied in Africa where they are maintained by specialist grazer mammals. The extinct grazers in Madagascar are assumed to have been hippos, giant tortoises, and, perhaps, some elephant birds. But isotopic evidence of the diet of sub-fossils is equivocal and has been interpreted as indicating lack of grass in the diets. How then, does one explain a whole suite of grass species, with many endemics, apparently adapted for heavy grazing pressure? This study is well designed and well implemented to show the distinct and divergent grass assemblages. There are clear differences in community assemblages and clear differences in the suite of functional traits of grass species. Species with short-spreading versus tall, caespitose architectures are sorted into short grass, heavily grazed versus tall grass frequently burnt assemblages. The phylogenetic analysis shows no deep phylogenetic conservatism so related taxa may occur in very different habitats. The authors make the interesting claim that the arrival of livestock 'rescued' the grazer-tolerant grasses from extirpation by more competitive tall grass species when the indigenous grazer guild was lost.

Though the study is impressive, the writing is quite turgid in places and probably overlong for Proc Roy Soc. The analytical methods are not standard so more care is needed in explaining simply and clearly what they show. The figures on my copy were largely illegible with very small fonts. The figure legends are uninformative as to what they represent and require a close reading of the manuscript to interpret. The paper deserves to be read by a wide readership, many of whom will not be botanically minded, so it is worth taking more effort over presentation.

Thank you for the feedback and comments. We agree that the writing was heavy going and clunky making it difficult for a reader to understand. We have re-written the methods and results for clarity and simplicity (lines 115 - 280). It is difficult to remove much text here as this is a complicated and interwoven set of analyses bringing together different methods and statistical tools to answer questions about the Malagasy grass flora. We hope that our amended text is easier to read and understand for a generalist audience.

The length of the methods section was increased by 217 words (to include a new description of our five functional traits). However, the improved methods enabled us to be more succinct in the results which we reduced in length by 125 words.

Further, the introduction is reduced in length by 68 words and the discussion by 63 words.

Further, we have amended figures to improve presentation and figure legends re-written to be more informative and clearer.

Minor comments.

There are a number of minor spelling and grammatical errors;

l.15, understating = understanding; l.78, arguments = argument; l.140 tared= tarred; l.226

Schizacyrium l.230. Crasperorhachis; l.316 Hemspson; l.346. Sentence beginning 'Given that' is not complete. L. 358. (Hempson et al 2015)? Replace ? with .

Thank you for these. We have corrected spelling and grammatical errors.

l.57-63. On diets of sub-fossils. The discovery of sub-fossils in the Central highlands by Samonds et al looks very promising for revealing more on diet of hippos. It is worth comparing their interpretation with that of isotopic studies of hippo diet in Africa such as Boisserie et al. 2005 *Palaeogeography, Palaeoclimatology, Palaeoecology*, 221(1-2), 153; and Cerling et al 2008, *J. Zoology*. These show that hippos consumed a lot of C3 material with their C4 grasses. Indeed African hippo isotopic signatures seem close to that of Samonds et al. Based on your grass data, more effort is surely needed on exploring sub-fossil diets.

Thank you for your suggestions. We have included these suggested references and raised similarities in isotopic signatures between Africa and Madagascar (lines 448 - 453). Isotopic values reported by Boisserie et al. (2005) and Cerling et al. (2008) for hippos indicate that modern and fossil African

hippos consumed C3 sedges and dicots with C4 grasses. Although Samonds et al. isotope data is lower compared to the African values, they also suggest a mixed C3 and C4 diet which can also be supported by the abundance of C3 forbs on grazing lawns.

I.120-127. You need to explain, very briefly, why these 'functional' traits matter here. The reader should not have to go to the supplementary files for this essential information.

Thank you for highlighting the oversight. We have added a brief descriptive paragraph about the function of each of the five traits within the main manuscript (lines 138 - 157).

I. 296. Human arrival dates seem odd. Intro gives 6000 BP as date of arrival.

We have clarified the reference on human arrival and used Anderson et al. (2018) 's date: 1350-1100 BP throughout the manuscript (lines 51 and 395). It is worth noting that dates for human arrival are still a subject of contention.

I.306. Is this the only reference on the need for heavy grazing to maintain short grass systems?

Surely earlier papers are available to strengthen your argument.

We have added McNaughton (1988), a pioneering work on grazing lawns as reference to how heavy grazing can maintain grazing lawns (lines 405).

I.339. I suggest reading Verweij et al. Oikos 2006 as a source of older literature on grazing lawns and a study where hippos maintain grazing lawns in high rainfall tall grass savannas.

Thank you. It is a great paper. We have added this reference to help with the discussion of how hippos can initiate grazing lawns in high rainfall areas (line 447).

I.366. 'grazing tortoises'. Grazing and browsing tortoises existed as inferred from presence or absence of a slot for raised neck.

Thank you for pointing this out. We have replaced "tortoises" by "grazing tortoises" (line 485).

I.379. Grasses can live for many centuries. However they can have rapid dynamic changes in dominance if conditions change.

We have reformulated the sentence to capture Reviewer 1's suggestion saying: "While grasses can

be long lived, it would be possible for grazing grasses in particular to be rapidly lost from ecosystems when over-topped by taller grasses or woody plants.” (lines 502 - 505).

Figures. The legends are obscure.

All Figure legends have been re-written for clarity (lines 544 - 590).

Figure 1. Correlations of what with what? My (naïve?) expectation is that the blue cells should be +ve correlations and the red correlations –ve.

Figure 1 shows correlation values indicating the likelihood pairwise species co-occurrence. We have amended the figure caption and added more detail about what is presented. Positive correlations are represented by blue cells and negative associations correspond to red cells (lines 546 - 555).

Figure 2. I could not make sense of this figure. Symbols were too small to read on x axes. What do the bars represent? Species? Plots? Figures should be easy to read without mining through the manuscript.

Figure 2 has been replaced with boxplots of model coefficients of environmental correlates for each assemblage group and the associated legend have been re-written for clarity and is now hopefully easy to understand.

Figure 3. What is ‘HCPC’? FGA, FGB, FGC are also very cryptic names for the three functional groups. Why not tell the reader ‘grazer’, ‘fire’ etc. in the legend or in the figure labels to aid interpretation of the data?

1) HCPC is now clearly defined as Hierarchical clustering on principal components on Line 565.

2) We have changed how we refer to groups following this suggestion. Throughout the manuscript, we have removed all reference to FGA, FGB and FGC and now use the terms “grazing group”, “fire group” and “intermediate group”. The initial acronyms were used so as not to interpret within the results section but changing these names has improved clarity and flow of the manuscript. We still refer to Assemblage 1 and Assemblage 2 in the results, as it is via the functional analyses that we determine which assemblages are related to fire or grazing.

Figure 4. Quite a poor fit between community group classification and functional group

classification. The problem seems to be greatest in the *Sporobolus* clade. Worth a comment?

Community (now assemblage) and functional group are not entirely convergent, but do have very strong overlap. However, given our methods we would not necessarily expect a 1:1 match. The main source of the mismatch is likely as a product of traits of species being collected once off in the study, and where a species was first encountered. Most species that are the source of the mismatch were first encountered in frequently burnt sites. If species are capable of tolerating, to some degree, both grazing and fire, we would expect those species to have a more plastic morphology and high intra-specific variability in traits such that if the species had first been encountered on a grazing lawn, then it is likely that those species could have been clustered into another group, perhaps the intermediate group. Only after the fact did this highlight to us the variable nature of the traits of species that can tolerate some level of both fire and grazing. We have gone on to amend our trait sampling protocols as result of this study, such that traits are now collected relative to the disturbance and region (to capture how other aspects of environmental variation could alter traits). Further, we would also expect diversity in grass functional types within Assemblages given that community composition shifts over time relative to environment and consumer controls.

*Definitely *Sporobolus pyramidalis* could be found in both fire and grazing-maintained communities. But there are important widespread species from other genera for which this is also the case, e.g. *Hyparrhenia rufa* and *Heteropogon contortus*.*

We have added a discussion about these points in lines 426 – 433.

In addition, we have added an analysis to test the variation of the measured trait across the phylogeny with a phylogenetic ANOVA, using the assemblage groups as predictor and found that leaf table height and bulk density variations were associated with assemblage. This supports the convergence between assemblage and functional groups that we found.

Supplementary figure 7 is worth placing in the main text to indicate the two communities. It is remarkable to me that the grazing lawn has endemic grass species from Madagascar and not just weedy species imported from Africa.

Thank you for the suggestion. We have moved supplementary figure 7 to the main text and refer to it in the discussion (now Figure 5).

Referee: 2

Comments to the Author(s)

This paper presents a coherent and well substantiated data set, convincingly analysed and

presented, that weighs in strongly to support an ancient vs anthropogenic origin for Malagasy grasslands. The combination of field collected data, analysed using functional type data and from a trait perspective, together with the phylogenetic evidence presents a strong case indeed. I would prefer to see the work of Bond et al 2008 highlighted a little more in the introduction as a vanguard of stimulating and adding valuable fuel to this debate, but I note that he was consulted and is acknowledged for this.

Thank you. We do refer to Bond et al 2008 in the introduction. It is worth noting that while this paper called into question the antiquity of Malagasy biome distributions, it focused on counting C4 grasses and diversity, which has also since been updated by Vorontsova et al 2016 and Hackel et al 2018.

In the discussion, these sentences are important for management implications, but not supported by a reference " Cattle, hippos and tortoises share key functional similarities, they prefer highly palatable grasses with high bulk density to maximise intake of nutritious food per bite. The replacement of one grazer with another is unlikely to have substantially reshaped diversity where an obligate grazing flora already existed." A reference here is important.

We now refer to McCauley et al. (2018) that did look, albeit indirectly, at the functional equivalence of cattle and hippos in maintaining grazing lawns in Africa (lines 487 - 489). The results demonstrated that a mixture of herbivores (including cattle and hippos) and removal of hippos on grazing lawns in East Africa similarly impacted the structure and diversity of grazing lawns.

Further, are the authors implying that conservation of the grassland diversity is not under threat, and I wonder about current efforts towards afforestation for the purposes of charcoal production and carbon sequestration. This might be worth mentioning if the authors feel it is relevant.

Definitely not. And, we are glad this point was raised. We have re-written added information in the last paragraph of the discussion on the extensive tree planting programs in the region where exotic species known as invasive elsewhere in the world are the most commonly planted. We elaborate and clarify threats and impacts to grassland systems on lines 501 - 522.

Appendix B

Associate Editor

Board Member

Comments to Author:

The manuscript has improved greatly through the response to the reviewers' comments. There are, however, a number of minor issues that have appeared into the manuscript as identified by Reviewer 1. These will need attention in another revision.

Thank you for the positive feedback on our re-submitted manuscript. We have fixed the issues identified by Reviewer 1.

Reviewer(s)' Comments to Author:

Referee: 1

Comments to the Author(s).

Thanks for the extensive edits and the helpful explanations of responses. The paper reads far better, the figures are easily interpreted now with the revised figure legends, and the edits to the discussion enhance the importance of the work. Unfortunately quite a few minor edits have crept in and should be corrected in a very minor revision. Here is my list.

Thank you for the positive feedback. In our attached manuscript we have corrected the identified mistakes.

Abstract.

Line 25 to 28. Minor edit suggested.

Replace: . "Within each assemblage, levels of endemism, diversity and grass ages support these as ancient assemblages, and where grazed dependent grasses co-evolved with the now-extinct megafauna, likely hippos and giant tortoises."

With: Within each assemblage, levels of endemism, diversity and grass ages support these as ancient assemblages. Grazer-dependent grasses co-evolved with the now-extinct megafauna, likely hippos and giant tortoises.

We have broken the sentence in two and replaced it with the suggested text (lines 25 – 28).

I. 45. Lehmann et al., 2011

We have corrected the citation to Lehmann et al., 2011 (line 46) and added the suggested papers (Cerling et al. 1997, Jacobs et al. 1999, Stromberg 2005, Edwards et al. 2010) in lines 86 – 87, providing paleo-evidence for grassland antiquity.

I. 46. Fix 'extinction alongside and the introduction of'

We have fixed the sentence and deleted the "and" after "alongside" (line 48).

Our authorship group has been in the process of writing a book chapter for the Natural History of Madagascar that has required a substantial review of palaeo evidence for human arrivals and evidence for fire and grazing. To that end, we have updated dates associated with first evidence of human arrival and the first evidence

of human landscape modification (lines 47 – 50, 299 – 301, 329 – 333). We have added these to the reference list (lines 524 – 527; 559 – 561).

I.57. delete 'grazers'

We have deleted "grazers" (line 60).

I.59-62. Fix this sentence. E.g. replace 'data support ' with 'data show that'

We have replaced "data support" with "data show that" as suggested and the sentence now reads "Existing isotopic data show that hippos and tortoises consumed primarily C₃ plants" (lines 62 - 63).

I.83. Well cited papers giving evidence for antiquity of grasslands (and especially savannas) include Cerling et al. 1997, Jacobs et al. 1999, Stromberg 2005, Edwards et al. 2010 along with Lehmann et al 2011.

We have added these well cited papers here (line 87) and also in lines 45 – 46. We have added them to the references list (lines 506 - 508, 524 - 527, 578 - 579, 647 - 649).

I.97. why not replace this line with just 'closed forest'. Some are on drainage lines others are not, e.g. Ambohitantely etc.

We agree that some forest is on drainage lines and others are not and have replaced "woody vegetation ecosystems in gallery forests along drainages lines" with just "closed forest" as suggested (line 101).

I.107. Fix 'relative frequency of in a uniform..'

We have fixed the sentence and deleted "of" (line 111).

I.132-133. Something missing here. From the Discussion, it would seem to be 'palatable whereas long thin leaves ignite easily and burn intensely'.

We have fixed the sentence by adding the missing phrase "long and narrow leaves" in front of "ignite easily and burn intensely" to read "Ratio of leaf width to leaf length reflects leaf shape with wide short leaves preferred by grazers as palatable and long narrow leaves ignite easily and burn intensely (Schwilk, 2015)" (lines 136 – 138).

I.270. Fix 'these evidence'

We have corrected "these evidence" to "the evidence" (line 275).

I.320. Delete 'a' at end of line

We have corrected the sentence as something was missing in it. Now it reads "In our dataset, these species were functionally clustered within the fire-grasses, but possibly as a product of traits being sampled where species were first encountered in our surveys, i.e., in frequently burnt communities, while these species were also found elsewhere." (lines 325 – 328).

I. 326. 'data suggest'

We have corrected "data suggests" to "data suggest" (line 335).

I.334. fix 'a mixed of C3 and C4'. Incidentally, check journal usage for whether C3 and C4 have subscripts.

We have corrected all C3 and C4 throughout the manuscript to have subscripts following the journal usage.

I.344 -345. Fix placing of commas to make sense of sentence.

We have added a coma after "We suggest" and deleted the one after "megagrazers" and the sentence now reads "We suggest, the ecology of the grasses examined here demonstrates that in the early Pliocene, megagrazers most likely hippos and giant tortoises were instrumental in the evolution and assembly of the Malagasy obligate grazing lawn flora" (lines 353 – 356).

We have made some minor edits to the manuscript to improve clarity and brevity (lines 27 – 28, 54 – 56, 62 – 65, 84 – 85, 88 – 92, 120, 123 – 124, 125, 141 – 143, 298 – 299, 315 – 317, 336 – 337, 342 – 343, 345 – 348, 350 – 353, 355 – 357, 358 – 359, 394, 396, 399 – 404, 449 – 454).